# Relationship of Mineralogical Composition to Thermal Expansion, Spectral Reflectance, and Physico-Mechanical Aspects of Commercial Ornamental Granitic Rocks

**DOI:** 10.3390/ma15062041

**Published:** 2022-03-10

**Authors:** Abdullah M. Alzahrani, El Saeed R. Lasheen, Mohammed A. Rashwan

**Affiliations:** 1Department of Civil Engineering, Engineering College, Taif University, Taif 21944, Saudi Arabia; amyalzahrani@tu.edu.sa; 2Geology Department, Faculty of Science, Al-Azhar University, Cairo P.O. Box 11884, Egypt; elsaeedlasheen@azhar.edu.eg; 3Geological Sciences Department, National Research Centre, 33 El BohoothSt. (Former El TahrirSt.), Dokki, Giza P.O. Box 12622, Egypt

**Keywords:** ornamental stones, granitic rocks, mineralogy, spectral reflectance, thermal expansion, physico-mechanical properties

## Abstract

The aim of the present study is to link the thermal expansion, spectral reflectance, and physico-mechanical aspects of different types of commercial granitic rocks with their mineralogical and chemical composition. The granitic rock types were characterized using several analyses, including petrography, chemical, mineralogical, and thermo-gravimetrical analysis using XRF, XRD, and TG/DTG/DSC techniques. The rock types were subjected to several performance tests, such as tests of their thermal expansion, spectral reflectance, and physico-mechanical properties. The results revealed that quartz, albite, and potash feldspar with minor amounts of mica (biotite/muscovite/annite) are the main mineralogical constitutes, in addition to some alteration minerals, such as kaolinite, saussorite, and prehnite. The studied granitic rocks were classified as monzogranite/syenogranite of high K-calc-alkaline and peraluminous characters and are related to late- to post-collisional settings. The thermogravimetrical analysis revealed that the overall mass loss over the whole temperature range up to 978 °C did not exceed 3% of the initial weight for all studied rocks. The results of thermal expansion revealed that the maximum change in linear thermal expansion for all rock types did not exceed 0.015% of their initial lengths at an unusual air temperature of 50 °C. The spectral analysis revealed that iron and hydroxyl ions are the main spectral absorption features that appeared in the VIS-NIR and SWIR regions, in addition to the appearance of the common and distinctive absorption peaks of the main mineral composition. Furthermore, the spectral reflectance demonstrated that the granitic rock types of low iron oxide content achieved a high reflectivity percent in the VIS-NIR and SWIR spectral regions compared with those of high iron content. As a general trend, the granitic rock types of high iron content and/or lower quartz content exhibited a high performance regarding their physical and mechanical properties, such as water absorption, apparent porosity, bulk density, compressive strength, and abrasion resistance. The results of water absorption, density, strength, and abrasion resistance of the studied granitic rocks are in the range of 0.14–0.31%, 2582–2644 kg/m^3^, 77.85–222.75 MPa, and 26.27–55.91 Ha, respectively, conforming to the requirements of ornamental stones according to the ASTM standard.

## 1. Introduction

Ornamental stones are natural stones of igneous, sedimentary, and metamorphic origins that can be cut into blocks and slabs and used for flooring, paving, cladding, funeral monuments, and statues due to the varieties of their colors in addition to their soundness and compactness. According to Gomes et al. [1], ornamental stones are subdivided into two main groups: the granites group, which includes all igneous rocks composed of quartz and feldspar minerals, and the marbles group, which includes all carbonaceous rocks composed mainly of calcite and dolomite minerals (such as marble). Marble and granite are the most famous types, but other stones such as quartzite, which is a metamorphic rock composed entirely of quartz, are also included [1].

According to the UN international classification of exchanged goods, the granite and marble definitions are intended on commercial terms [2], where the “Marble” term comprises the whole class of carbonate rocks amenable to sawing and polishing, thus going beyond the limits of the mere lithologic characterization, while the “Granite” term includes the whole set of eruptive igneous rocks having a granular structure with a poly-mineral composition, irrespective of the content of quartz.

Based on the USGS report of 2013, the worldwide production of ornamental stones in 2012 was estimated at about 125 million tons [3]. The increase in world production in 2018, as reported by Montani [4], reached about 152 million tonnes. These ornamental stones were produced by twenty-seven countries with various share percentages. The top ten countries that share more than ninety percent of the international production of ornamental stones are ordered as follows: China, India, Turkey, Brazil, Iran, Italy, Egypt, Spain, USA, and Portugal [4]. Due to the high distribution of natural stones such as granitic rocks and sandstones, they have been widely used as construction and building materials in historical and cultural heritage [5].

According to Montani [4], it was noted that Egypt occupies the seventh rank in the production of ornamental stones globally, with about a 3.6% (5.25 million tons) share. In Egypt, more than fifty brands of ornamental stones of different rock types are produced.

In the present work, the authors focused on studying and evaluating several types of igneous rocks (commercially called granite rocks) from different localities in the Eastern Desert of Egypt as ornamental stones.

Geologically, Egyptian granitic intrusions are widely distributed in the northern sector of the Eastern Desert, which represents the western part of the Arabian Nubian Shield (ANS). These rocks are of different ages and mineralogical and geochemical compositions; furthermore, they are also of different tectonic regimes, ranging from Syn-orogenic (older rocks, 850–610 Ma) to late post-orogenic (younger rocks, 610–550 Ma) [6,7].

The characteristics and behaviors of the different types of granitic rocks as a function of variable mineralogical composition were studied by several researchers, as described below.

Lasheen et al. [7] studied the relationship between the natural radioactivity of some commercial granitic rocks and their petrographical description, including mineralogical constitutions. They found that the rock types with a lack of accessory minerals such as allanite, monazite, titanite, thorite, and zircon yielded low activity concentrations of natural radioactivity.

Ramana and Sarma [8], in their study on the thermal expansion of granitic rocks, confirmed that the linear thermal expansion coefficient is a function of several parameters, including the rate of heating, crack porosity, thermal cycling, mineral composition, and grain orientation. Janio de Castro Lima and Paraguassu [9] studied the linear thermal expansion of granitic rocks and found that the coefficients of thermal expansion increase with increasing quartz content while decreasing with increasing apparent porosity. According to Siegesmundet al. [10], the expansion coefficients of the individual rock-forming minerals in addition to the rock fabric are the main factors affecting the thermal expansion characteristics of a rock due to the variation in the dilatation behavior of minerals. Plevova et al. [11] used several thermal analytical techniques such as thermo-mechanical, thermogravimetric, and differential thermal analysis for studying the behavior of the thermal expansion of granites of variable mineralogical composition. They found a similarity in the thermal expansion values between the different granites and a little difference in their thermo-mechanical characteristics. This slight difference was attributed to the variation in their mineralogical content. Other authors studied the relationship between the spectral characteristics of different types of igneous rocks and their mineral composition [12,13,14,15,16].

Ludovico-Marques et al. [5] studied the mechanical evaluation of some historical building stones, particularly granite and sandstone, using physical properties. They found, using a proposed model, a distinct dependency of the mechanical properties of the rocks on their physical properties, particularly porosity.

The textural features, mineral composition, physical and mechanical properties, and the exposing environmental conditions are considered the important factors affecting the alteration and deterioration of the natural and artificial stone materials [17].

The research significance of the present paper is as follows.

The most pronounced characteristic of the granitic igneous rocks is the great variation of their mineralogical composition and, consequently, their chemical composition. Such variations in mineralogical composition could have a great effect on their physical properties, particularly density, as reported by Siegesmudet al. [10] and Gautamet al. [18]. Furthermore, the variation in the mineral composition of igneous rocks may affect their thermal and spectral reflectance behaviors [8,9,10,11,12,13,14,15,16].

Moreover, looking for the literature review, there are not enough studies about the physical and mechanical properties of granitic igneous rock and their relationship to the mineralogical composition and their suitability as ornamental stones.

Therefore, the present work is a comprehensive study that aims at linking the variations in thermal expansion behaviors, spectral reflectance features, and physico-mechanical properties of different types of commercial granitic rocks to their mineralogical and chemical composition.

## 2. Materials and Methods

### 2.1. Materials (Granitic Rock Types)

Eight types of granitic rocks of commercial names (Gandonna, pink granite, buff granite I (coarse grain), buff granite II (medium grain), red granite, Fantazia, Rosa granite, Qusseir brown) used as ornamental stones were selected in this study (Figure 1). These commercial types were supplied from the marble and granite factories located in Shaq El-Theban Area, Cairo.

### 2.2. Methods

The aim of the present study was accomplished by characterizing the rock types from petrographical, mineralogical, chemical, and thermogravimetrical points of view using a PLM (polarizing light microscope), X-ray diffraction (XRD), X-ray fluorescence (XRF), thermogravimetry (TG) and its derivative (DTG), and differential scanning calorimetry (DSC) techniques, respectively. In addition, the thermal expansion, spectral reflectance, physical properties (water absorption, density, and porosity), and mechanical properties (compressive strength and abrasion resistance) were also measured. The results of the assessment of the physico-mechanical properties are compared with the international standard specifications related to ornamental stones.

#### 2.2.1. Petrographical Investigation

The rock types were prepared by cutting the rock samples into thin slabs and polishing them carefully into thin sections (one section per rock type). The thin sections were then petrographically investigated under a polarizing light microscope (Olympus bx53 type (Tokyo, Japan)).

#### 2.2.2. X-ray Diffraction Analysis

The selected types of granitic rocks were crushed and ground to a fine powder shape (<63 μm) using a ball mill machine. The ground rocks were quartered several times until the representative quantity was selected (one sample per rock type). The selected quantity was analyzed for mineralogical composition in the range of 4 to 75 (2θ) using XRD (Philips 1730 diffractometer with Ni filter, at a scan speed of 0.5 mine1, Cu Ka radiation, National Research Centre, Dokki, Egypt). The quantitative analysis of the rock-forming minerals was also performed using Reitveld refinement.

#### 2.2.3. X-ray Fluorescence Analysis

The full chemical composition was assessed, including major oxides (%) and minor elements (ppm) in addition to the loss on ignition (L.O.I., %) of the ground rocks of less than 63 μm size (2 samples per rock type). The samples were measured using XRF at the National Research Centre Laboratories (Axios PANalytical 2005, Sequential WD-XRF Spectrotometer, Almelo, The Netherlands, using ASTM E-1621 standard guide for elemental analysis by wavelength-dispersive X-ray fluorescence spectrometer). The data were processed through the advanced data treatment software of the PAnalytical Super Q package with the multi-elemental synthetic standards prepared by the BGS/PAnalytical Corporation.

#### 2.2.4. Thermo-Gravimetric Analysis

The thermal behavior of the selected rock types under gradual elevating, temperatures expressed by thermogravimetry (TG), derivative thermogravimetry (dTG), and differential scanning calorimetery (DSC) was measured. The powdered rock types (<63 μm size) of 16 mg mass were subjected to heating at 10 °C/min over the whole temperature range of 34 to 978 °C under inert nitrogen gas conditions using (LabSys EVO Setaram Instrumentation, Caluire-et-Cuire, France).

#### 2.2.5. Linear Thermal Expansion

The measurement of the linear thermal expansion coefficient (αL) and thermal strain (ΔL) for the selected granitic rock types was carried out using a dilatometer (model NETZSCH DIL 402 PC (Selb, Germany)). The rock types were prepared in a rod shape of 12.5 mm length (one sample per rock type) and heated at a rate of 5 °C/min in a temperature range of 30 to 1000 °C.

#### 2.2.6. Spectral Reflectance

The spectral reflectance features of the rock types were measured using a double-beam spectrophotometer (GascoCorp., V. 570, Rev. 1, Easton, PA, USA) in the wavelength range of 350 to 2500 nm at intervals of 2 nm, with 1156 total points. The rock types were prepared in a polished square shape of 2 cm × 2 cm area and 0.5 cm thickness.

#### 2.2.7. Physico-Mechanical Properties

The selected rock types were prepared as cubic shapes of 50 mm × 50 mm × 50 mm dimensions using a rock-cutting machine. The physical properties (water absorption, apparent porosity, and bulk density) were measured for the prepared rock types (six samples per test) according to standard test methods [19,20] at room temperature. The mechanical properties (compressive strength and abrasion resistance) were measured according to standard test methods [21,22].

## 3. Results

### 3.1. Petrographical Investigation

An Olympus bx53 polarizing microscope was used for petrographic studies in order to identify the mineralogical constituents as well as textural relationships for each commercial rock type. The petrographical properties of the granitic rock types were presented in Figure 2 and summarized in Table 1. According to their modal analysis (quartz-plagioclase-K-feldspar percentage), the examined commercial rocks were classified based upon IUGS classification [23] and described as follows:

The Gandonna rock type is a whitish-grey color and is characterized as a medium- to coarse-grained hand specimen (Figure 1a). Petrographically, this rock type is described as a “monzogranite”. It is composed of plagioclase, K-feldspar, quartz, and biotite as essential minerals. Allanite and zircon are the main accessory minerals. K-feldspar is represented by anhedral, medium- to coarse-grained (up to 6 mm in length and 4.3 mm in width), flamy (veins) orthoclase perthite and patchy perthite. Patchy perthite exhibits slight turbidity, whereas orthoclase perthite is pristine (Figure 2a). Quartz occurs as medium- to coarse-grained and exhibits a normal extinction. Plagioclase is represented by subhedral and tabular crystals. They include two types of plagioclase: the first is medium-grained, reaches up to 2.7 mm in width and 4.2 mm in length, is commonly Carlsbad-zoned and shows extensive saussuritization, while the second type is fine-grained, up to 1.2 mm in length and 0.4 mm in width, commonly reveals lamellar twinning and is less altered. Biotite occurs as flaky crystals mostly altered to chlorite and associated with plagioclase. Allanite occurs as reddish-brown crystals associated with biotite crystals. It is fractured and filled by iron oxides (Figure 2b). Short minute crystals of zircon occur as fine-grained and embedded in orthoclase perthite.

The pink granite rock type is characterized by a reddish color with a medium-grained in the hand specimen (Figure 1b). Petrographically, this rock type is classified as “monzogranites”. It is composed of plagioclase, K-feldspar, and quartz as the main constituents. Potash feldspars occur as medium-grained and turbid perthite and antiperthite. String and patch are the two types of perthite, while antiperthite is of patch type (up to 4.3 mm length × 4.1 mm width). A reaction rim of albite occurs between two crystals of perthite. Plagioclase occurs as medium-grained and is extensively saussuritized. In addition, lamellar twinning, pericline as well as zoned crystals occur (Figure 2c). Quartz mostly occurs as fine- to medium-grained and anhedral crystals. Normal as well as undulose extinction occur. Biotite occurs as fine-grained, flaky crystals that are partially altered to chlorite along their peripheries. Zircon and titanite are the main accessory minerals. Zircon is present as a euhedral that is commonly embedded in plagioclase crystals. Titanite is common as subhedral to euhedral (sphenoidal shape) (Figure 2d), is usually associated with biotite and quartz, and occurrs as aggregates.

The buff granite “I” rock type (coarse-grained) is described as coarse-grained with a buff color in the hand specimen (Figure 1c). Petrographically, this rock type is classified as “Syenogranite”. It mainly comprises potash feldspars, plagioclase, quartz, and muscovite minerals. K-feldspars are represented by coarse-grained perthite and microcline, which exhibit slightly turbid surfaces (due to the development of kaolinitization). Occasionally, these rocks engulf quartz, plagioclase, and muscovite crystals. Plagioclase (albite) occurs as medium-grained, euhedral to subhedral crystals. It exhibits lamellar twinning with slight saussuritization. They are sometimes twisted and fractured as a result of deformation processes (Figure 2e). Quartz occurs as fine to medium-grained, anhedral crystals, rarely showing undulose extinction. Muscovite flacks occur as fine- to medium-grained crystals and are found either as primary or secondary shreds filling fractures of potash feldspar.

The buff granite “II” rock type (medium-grained) is described as medium-grained with a buff color in the hand specimen (Figure 1d). Petrographically, this rock type is classified as “Syenogranite” and consists mainly of K-feldspar, plagioclase, quartz, and muscovite as essential minerals. K-feldspars are represented by anhedralflamy and patchy sections. They mostly engulf plagioclase and muscovite crystals (Figure 2f). Plagioclase occurs assubhedral, tabular crystals measuring 3.5 mm in length and 1.8 mm in width. It shows lamellar and pericline twinning and is rarely zoned. It reveals slight to extensive saussuritization (Figure 2g). Quartz occurs as fine- to medium-grained, anhedral crystals, and corrodes other constituents. Irregular flakes of muscovite are commonly associated with perthite.

The Fantazia rock typeis characterized by a white color with a medium- to coarse-grained in hand specimen (Figure 1e). Petrographically, this rock type is classified as “monzogranite”. It consists essentially of albitic plagioclase and quartz with a small amount of K-feldspar. Albite crystals are tabular, pristine, and medium-grained (Figure 2h). They show both pericline and lamellar twinning that is slightly altered to saussurite. Occasionally, albite is twisted, fractured and cut by quartz. Quartz occurs as fine- to medium-grained, with normal extinction and mostly corroded plagioclase crystals. K-feldspars occur as patchy, pristine perthite. Biotite occurs as a fine-grained, flaky crystal and is hosted by plagioclase crystals.

The red granite rock type is described as a medium grain with a reddish color in the hand specimen (Figure 1f). Petrographically, this rock type is classified as “Syenogranite”. It constitutes mainly of K-feldspar, quartz, plagioclase and biotite. Potash feldspar is represented by extensive turbid crystals, medium to coarse-grained (2.7 mm length × 3.1 mm width) perthite (Figure 2i,j), orthoclase perthite and antipertite crystals. They suffered from high kaolinization processes. Quartz occurs as medium-grained, anhedral (angular) to subrounded crystals with normal extinction and wavy extinction. It is commonly cracked and surrounded by perthite. Plagioclase is represented by subhedral, tabular, medium-grained (1.6 mm length × 1.2 mm width) crystals and exhibitspericline and lamellar twinning that has sometimes disappeared due to the kaolinization process. A small amount of biotite occurs as fine-grained and is completely altered to chlorite and muscovite along their cleavage and peripheries.

The Rosa granite rock typeis a pinkish-white color and is characterized by a medium- to coarse-grained in-hand specimen (Figure 1g). Petrographically, this rock type is classified as “monzogranite”. It contains K-feldspar, quartz, plagioclase, and biotite as essential minerals. K-feldspar is represented by pristine, medium to coarse-grained orthoclase perthite. Quartz occurs as medium-grained with normal extinction. Plagioclase is represented by tabular crystals that show clear lamellar, pericline and zoned twinning. Fine- grained biotite occurs as flaky crystals, mostly altered to chlorite and mostly enclosed in orthoclase perthite.

The Qusseir brown rock type is a brownish color and is characterized by a medium-grained in-hand specimen (Figure 1h). Petrographically, this rock type is classified as “monzogranite”. It is equigranular, medium-grained and composed essentially of potash feldspars, plagioclase, quartz, and biotite. Potash feldspar is represented by turbid, fine- to medium-grained (av. 2.9 mm length × 2.7 mm width) perthite crystals. They are corroded by quartz, engulfing the skeletal quartz crystal, and are fractured and filled by quartz. Quartz occurs as interstitial grains filling the fractures between other constituents. It commonly reveals undulose extinction (Figure 2k). Granophyric textures are common due to intergrowth between quartz and potash feldspar (Figure 2l). Turbid plagioclase occurs as tabular, medium-grained crystals that are completely kaolinitized and occasionally fractured and filled by quartz. Biotite is represented by one crystal, which was fine- grained and surrounded by perthite crystals.

### 3.2. Geochemistry

The chemical analysis (major (%) and trace elements (ppm)) of all commercial granitic rock types are listed in Table 2. The studied granitic rocks have high contents of SiO_2_ (69.7–76.2%) and total alkalis (7.6–10.79%), as well as significant concentrations of Al_2_O_3_ (12.74–15.76%). In addition, they possess low (less than unity) concentrations, on average, of TiO_2_ (0.16%), MgO (0.36%), CaO (0.88%), P_2_O_5_ (0.09%) and loss in ignition (0.46%). Moreover, they have low contents of Mg# (5.26–34.36) and K_2_O/Na_2_O (0.45–1.05).

According to Cox et al. [24], all granitic rock types fall in the granitic field (Figure 3a). These rock types range from granites (Gandonna, pink granite, Fantazia and Rosa granites) to alkali-granites (red granites, Qusseir brown, buff granites I and II) (Figure 3b) using R1-R2 according to De la Roche et al. [25]. The examined rock types have an alumina saturation index (ASI) that varies from 1.2 to 1.5, reflecting their peraluminous signature (Figure 3c). Moreover, their agpaitic index (AI) ranges from 0.55 to 0.73, indicating that they have calc-alkaline and peraluminous affinity [26]. They have high K calc-alkaline according to Rickwood, [27], except for Fantazia, which has medium calc-alkaline (Figure 3d). The examined granitic rocks are of I-type granites (Figure 4a) according to Hine et al. [28].

It is noticeable that there are wide variations and high concentrations in incompatible trace element contents, where Sr (8.4–158 ppm), Rb (70.2–282 ppm), Ba (6–621 ppm), and Zr (36–627 ppm) are relative to compatible elements such as 2–13.2 ppm, and 2.3–44.5 ppm. This suggests different fractional crystallization during their formations. Their trace element are normalized to a spider-pattern primitive mantle that exhibits Ba-, Nb-, Sr- and Ti-negative anomalies (Figure 4b).Tectonic discrimination diagrams can be used to manifest the tectonic regime for the studied Egyptian granitic rocks. The examined rocks have geochemical characteristics consistent with a late- to post-collisional setting (Figure 4c,d), as indicated by Pearce et al. [30] and Hassan and Hashad [32].

### 3.3. Mineralogical Composition (XRD)

As shown in Figure 5, the mineralogical analysis of the studied granitic rock types using X-ray diffraction revealed that quartz (SiO_2_, reference codes (01-083-0539; 01-070-7344; 01-089-1961; 01-089-8936)),plagioclase (albite: NaAlSi_3_O_8_, reference codes (98-008-7656; 01-072-8434; 98-008-7660; 98-003-4870)), alkali-feldspar (orthoclase: KAlSi_3_O_8_, reference code (98-003-1134)), and mica (annite: K (Fe Al) (Al Si) O_10_ (OH)_2_, reference code (01-074-2572)) are the main common minerals in the studied rock types. All these minerals were detected in various percentages as given in Table 3 based on their chemical composition (Table 2); for example, the high percentage of albite mineral in the granitic rock type (e) Fantazia was matched with the high presence of Na_2_O: 6.05%. The X-ray investigation also revealed a presence of alterations in some rock types, such as (a) Gandonna, (b) Pink granite, and (h) Qusseir brown. Such alterations are indicated by the existence of secondary minerals such as zeolite (Na_2_Al_2_Si_3_O_10_.2H_2_O, reference code (01-085-1554)), prehnite (Ca_2_Al_2_Si_3_O_10_.(OH)_2_, reference code (98-016-4246)) and kaolinite (Al_2_Si_2_O_5_(OH)_4_, reference code (01-075-1593)).

### 3.4. Thermal Analysis by (TG/DTG/DSC)

Thermogravimetric analysis (TG) is an analytical technique used to measure the thermal stability of a material and its fraction of volatile components by monitoring its mass change upon heating at a constant rate [33]. According to Gautam et al. [34], high temperatures have a very important effect on the physical and mechanical properties of rocks in the field of rock mechanics for several engineering applications, including, for example, underground nuclear waste storage. These effects are closely related to their mineral composition and microstructure features.

The thermogravimetry (TG) and differential thermogravimetry (DTG) curves, as graphically plotted in Figure 6, showed a variation in the mass as a function of temperatures. A significant mass loss was observed at various temperatures less than 200 °C for all rock types that represented the evaporation of physically (absorbed) and chemically bonded water that was liberated at 300 °C [34,35,36,37]. In this study, the reduction in mass because of the release of physically free (absorbed) and chemically bound water up to 200 °C was found to be 1.949%, 1.210%, 1.093%, 0.164%, 0.471%, 0.257% for Gandonna (a), pink granite (b), buff granite II (d), fantazia (e), rosa granite (g), and qusseir granite (h) at temperatures of 137 °C, 150 °C, 137 °C, 85 °C, 132 °C, and 96 °C, respectively. The rock types buff granite I (c) and red granite (f) showed an inconsiderable mass loss at that range of temperatures, which was confirmed by their thermo-gravimetric derivative peaks at the same figure that indicated a very low rate of mass change by time (%/min.). A further mass loss was observed with excessive heating between 400 and 800 °C. Such loss in mass was attributed to the releasing of crystal water [36,37] or water-bearing mineral phases [34].

The variations in mass loss of the different rock types may be attributed to their mineralogical composition, especially the secondary (alteration) minerals. It was noticed that the loss in mass in the rock types (Figure 6a,b,d) was higher than other types due to the presence of high percentages of alteration minerals. However, the overall loss in mass over the whole temperature range of 34 to 978 °C did not exceed 3% of the initial weight for all studied rocks. As shown on the differential scanning calorimetery (DSC) or heat flow curve in Figure 6, a small endothermic peak marked by a circle was detected around 570 °C for all rock types representing the phase transition of α-β quartz at 573 °C [11].

### 3.5. Linear Thermal Expansion

The relative change in sample length depending on the considered temperature interval is expressed as the linear thermal expansion [18]. The relationship between the coefficients of linear thermal expansion of different granitic rock types over a wide range of temperatures from 30 to 1000 °C was plotted in (Figure 7). In this figure, three relationships were plotted with the temperature change:(a) change in length or thermal strain (*dL*/*Lo*), (b) the linear thermal expansion coefficient (*α_T_*), and (*c*) the rate of linear thermal expansion (*dα*). It is well known that the coefficient of linear thermal expansion (*α_T_*) is calculated from the following equation [38]:$$\alpha_{T}=\left(dL/Lo\right)\times \left(1/dT\right)$$
where *dL* is the change in sample length (mm) over a temperature range (*dT*) (°C), *Lo* is the initial sample length (mm), and *dL/Lo* is the derivative of thermal strain. Table 3 shows the measurements of thermal strain (relative length change, %) and the coefficient of linear thermal expansion for the studied rock types at different temperatures.

As shown in Figure 7, the first sharp increase in the linear thermal expansion coefficient of the studied igneous rocks was detected around 570 °C, which represented the quartz mineral (α-β transition phase) that appears at 573 °C [39]. This phase is supported by the appearance or jump in the peak of the thermal coefficient rate (*dα*) in the same figure; in addition, it is well confirmed in the curve of differential scanning calorimetry (DSC) (Figure 6). A second increase or jump in the coefficient of linear thermal expansion was observed in the range of 700 to 900 °C for some rock types ((a) Gandonna, (b) pink granite, (c) buff granite I, (d) buff granite II). Such an increase is also supported by a second appearance or jump in the peak of thermal coefficient rate (*dα*) in the same figure, which may be related to the high percentage of quartz in their composition (Table 3). It could be attributed to the phase transition of quartz to hexagonal tridymite at about 800 °C [39]; however, this transition was not indicated in the differential scanning calorimetry (DSC) curve. As given in Table 4, an increase in the values of the linear thermal expansion coefficient was clearly noticed with increasing temperature until 700 °C; after that, it begins to decrease with temperature increases up to 1000 °C. It was also observed that the relative change in length for all rock types of different colors did not exceed 0.015% of the initial sample length at an unusual air temperature of 50 °C.

As presented in Table 4, it was observed that the linear thermal expansion coefficient up to 300 °C was 10.97, 10.95, 13.11, 13.58, 11.34, 14.7, 14.2, and 11.7 × 10^−6^/K^−1^ for the granitic rock types gandonna, pink granite, buff granite I, buff granite II, fantazia, red granite, rosa granite, and qusseir granite, respectively. These values seemed similar to that obtained by Gautamet al. [18] for various granitic rocks subjected to thermal expansion up to 250 °C.

### 3.6. Spectral Reflectance

As the rocks are composed of minerals and are compositionally more complex and variable than minerals, their diagnostic spectral curves are therefore difficult to define. However, the description of the spectral characters of rocks is possible based on the spectral characters of the constituent minerals and textural properties such as grain size; i.e., it depends on their chemical composition, mineralogy, grain size, and color [12].

In the present study, the spectral characterization of the different granitic rock types was determined in the solar reflectance region over the wavelength range of 0.35–2.500 μm or 350–2500 nm. This region is subdivided into the visible-near infrared region (VIS–NIR) in the range of 0.4–1.0 μm or 400–1000 nm and the short-wave infrared region (SWIR) in the range of 1.0–3.0 μm or 1000–3000 nm (Figure 8).

The results of bidirectional spectral reflectance of the eight igneous rock types (gandonna, pink granite, buff granite I, buff granite II, fantazia, red granite, rosa granite, gusseir brown) are graphically plotted against the wavelength in Figure 9a–h, respectively. It is observed from Figure 9 that the reflectance scale (*Y*-axis) for all rock types is not uniform due to the variable response of these rocks to spectral reflectance, which indicates the variation in the mineral content and type.

The rock types revealed a notable increase in the spectral reflectance with variable proportions from shorter to longer wavelengths in the visible-near infrared region (VIS–NIR) around 800 nm, except for rock type Fantazia (e), which exhibited an early increase in the spectral reflectance at lower wavelengths around 450 nm which decreased gradually with increasing wavelength. A higher spectral reflectance in the visible-near infrared region (VIS–NIR) was recorded for the rock type (e) fantazia, with increasesof about 67% observed. This was followed by rock types (d) buff granite II, (c) buff granite I, (b) pink granite, (g) rosa granite, (a) gandonna, (f) red granite, and (h) qusseir brown, with reflectance values of 57%, 50%, 46%, 40%, 35%, 28%, and 21%, respectively. This order is well-matched with the gradient in the colours of rock type, beginning with a light colour as in fantazia type (e) and ending with a dark colour as in qusseir brown type (h). This variation may be attributed to the variation in the iron content, as given in Table 2. The reflectivity values of the studied granitic rocks were deemed similar to those recorded by Bailin and Xingli [12] (40–70%) for several acidic rock samples.

It was mentioned that the different granitic rock types have similar mineral compositions as detected from X-ray diffraction analysis (XRD), with their compositions including quartz, plagioclase (albite), alkali feldspar (orthoclase), and mica (annite, biotite), while some alteration minerals such as kaolinite, zeolite, and prehnite were detected in some rock types. The double beam spectrophotometer analysis revealed the appearance of major common absorption peaks of different depths for all rock types at approximately similar wavelengths (400 nm, 900 nm, 1400 nm, 1900 nm, 2200 nm) that are distinctive to their mineral composition, as shown in Figure 9. The spectral curves of the studied rock types were found near to the standard characteristics features of the rock-forming minerals according to the library of the U.S Geological Survey (USGS, file report) [40].

It was observed from Figure 9 that the main spectral absorption features that dominated in the VIS-NIR (400–1000 nm) region are related to iron ions (Fe^2+^, Fe^3+^) [14]. Iron occurs as a variety of oxides and hydroxides, with ferric ions displaying a positive absorption border between 400–600 nm and two other bands near 520 nm and 900 nm. Moreover, ferrous ions display an absorption band near 1100 nm [12].

In the present study, two major broad absorption bands were detected for iron in the VIS-NIR region. One absorption band was around 400 nm, while the other band is around 900 nm.

Regarding the SWIR region of the wavelength range 1000–2500 nm, it is marked by the spectral absorption features of hydroxyl ions (O-H), water molecules (H_2_O), and carbonates (CO_3_) [12].

In the rock types Gandonna (a) and Pink granite (b), the absorption bands at the approximate wavelengths of 1400 nm and 1900 nm may be attributed to H_2_O molecules due to the presence of zeolite in their mineral composition according to XRD analysis. The hydroxyl ion (O-H) is a widespread constituent in rock-forming minerals such as clays, mica, and chlorites. In the present study, the spectral absorption features of (O-H) were observed at several bands around 1400 nm, 1900 nm, 2200 nm, which was typically for Al-OH, 2350 nm, which was typically for Mg-OH, and 2400 nm. Similarly, Zhou and Wang [15] recorded the presence of hydroxyl absorption bands for their studied granitic rocks at the same wavelengths, which indicatedthe existence of multiple hydrated mineral species. The presence of the last three bands is very common in clay and hydrous silicates. This is the case of rock types (a) gandonna and (h) qusseir brown due to the presence of kaolinite and annite minerals.

Figure 10 illustrates the relationship between the spectral reflectance of the rock types and their contents of iron oxide (Fe_2_O_3_, %). It was observed that the percentage of reflectance decreases with increasing iron content. These results were similar to those obtained by Borisova [41].

### 3.7. Physico-Mechanical Properties

Figure 11a–d shows the results of water absorption, apparent porosity, bulk density, compressive strength, and abrasion resistance of the studied granitic rock types. It was observed that the increase in the apparent porosity led to an increase in water absorption (Figure 11a) and, consequently, a reduction in dry and SSD bulk density (Figure 11b). The results revealed that the density of the studied granitic rock types ranged from 2582 kg/m^3^ to 2645 kg/m^3^. Siegesmund et al. [10] found that the values of apparent density and porosity of several granitic rocks are in the range of 2.54–2.67 g/cm^3^ and 0.3–3.37%, which seemed similar to the results obtained in the present study. It was observed that the rock types with high iron contents have a high bulk density, except for the rock type (e) fantazia, which was of a very low iron content (0.486%) but exhibited a higher bulk density (2619 kg/m^3^) compared to the rock type (f) red granite, which was of a higher iron content (1.126%). This is due to the lower apparent porosity of fantazia (0.54%) compared to red granite (0.66%).

According to the results of the compressive strength test, as plotted in Figure 11c, it was observed that the rock types of higher bulk density have higher compressive strength values except for the rock type (f) red granite, which demonstrated high compressive strength despite its relatively lower density. This may be attributed to its lower quartz content.

In the same line, the results of abrasion resistance, as plotted in Figure 11d, showed an increase in the groove width representing a reduction in the abrasion resistance. On the contrary, a decrease in the hardness of the granitic rock types occurs with decreasing bulk density.

Comparing the results of physical properties with the limits of international standard specification ASTM C615/C615M [42] relating to “Granite Dimension Stone”, it was found that the water absorption values of the granitic rock types range from 0.14% to 0.31%, which complies with the requirements of the water absorption limit (0.4% max.). The results of bulk density of the granitic rock types range from 2582 kg/m^3^ to 2645 kg/m^3^, which complies with the requirements of the density limit (2560 kg/m^3^ min.).

Regarding the results of the mechanical properties and their relationship with the standard specification (ASTM C615/C615M) [42], it was observed that the values of compressive strength of the granitic rock types are in the range of 77.85 MPa to 222.75 MPa, which indicates that not all rock types achieved the requirement of compressive strength limit (131 MPa min.). However, the rock types that did not satisfy the recommended limit can be used in building cladding, which does not require high strengths; they can also be used in light-duty purposes.

The values of abrasion resistance of all rock types range from 25.12 Ha to 55.91 Ha, which satisfies the requirement of the specification limit (25 Ha min).

## 4. Conclusions

Eight commercially named granitic rocks (gandonna, pink granite, buff granite I (coarse grain), buff granite II (medium grain), fantazia, red granite, rosa granite, and qusseir Brown) used as ornamental stones were petrographically described as monzogranites, monzogranites, syenogranites, syenogranites, monzogranites, syenogranites, syenogranites, and monzogranites, respectively, based upon IUGS classification. These rock types were classified as high K-calc-alkaline, I-type and peraluminous characters and are related to late- to post-collisional settings.

The X-ray diffraction analysis and petrographical investigation demonstrated that the main mineralogical constitutes are quartz, albite, and potash feldspar, with minor amounts of mica (biotite/muscovite/annite). Some alteration minerals such as kaolinite, saussorite, and prehnite, in addition to some accessory minerals such as zircon and titanite, were detected.

According to thermogravimetrical analysis, the overall mass loss over the whole temperature range up to 978 °C did not exceed 3% of the initial weight for all studied rocks. Moreover, significant mass loss was observed at various temperatures less than 200 °C for all rock types, which represented the evaporation of physically (absorbed) and chemically bonded water that was liberated at 300 °C.

The results of thermal expansion revealed that the maximum change in linear thermal expansion for all rock types did not exceed 0.015% of their initial lengths at an unusual air temperature of 50 °C.

The spectral analysis revealed that iron and hydroxyl ions are the main spectral absorption features appearing in the VIS-NIR and SWIR regions, in addition to the appearance of the common and distinctive absorption peaks of the main mineral composition. Furthermore, the spectral reflectance demonstrated that the granitic rock types with low iron oxide contents achieved a high reflectivity percent in the VIS-NIR and SWIRspectral regions compared to rock types with high iron contents.

The results of water absorption, density, strength, and abrasion resistance of the studied granitic rocks are in the range of 0.14–0.31%, 2582–2644 kg/m^3^, 77.85–222.75 MPa, and 26.27–55.91 Ha, respectively, conforming to the requirements of ornamental stones according to the ASTM standard.

## Figures and Tables

**Figure 1 materials-15-02041-f001:**
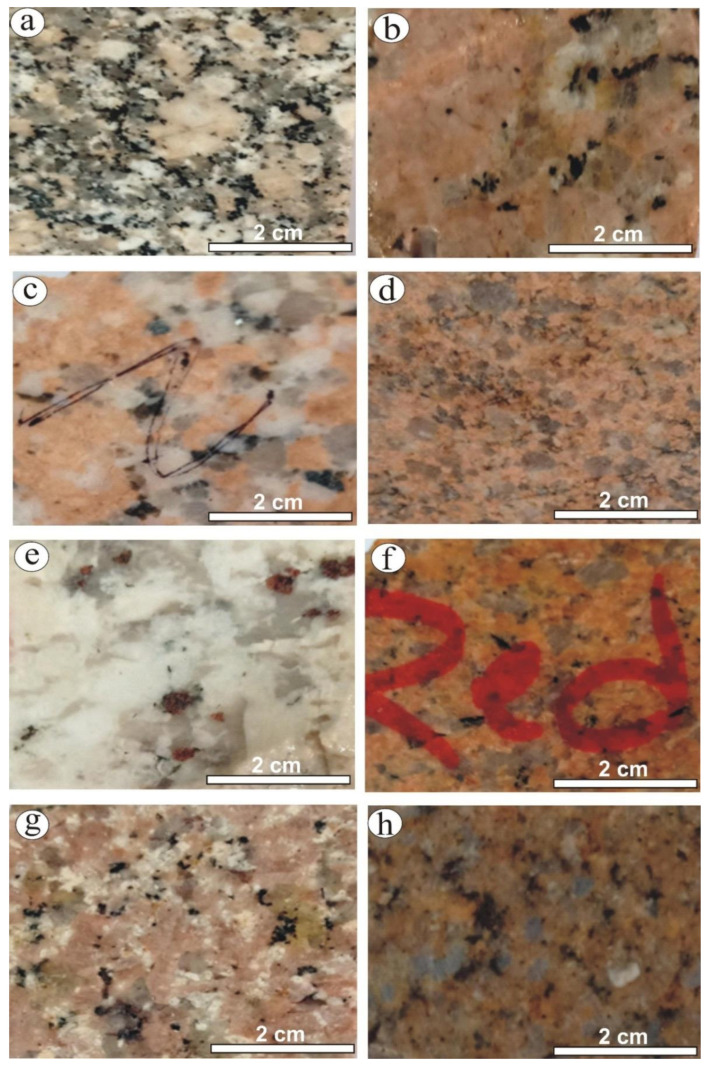
Photographs showing hand specimens of the granitic rock types: (**a**) Gandonna, (**b**) pink granite, (**c**) buff granite I, (**d**) buff granite II, (**e**) Fantazia, (**f**) red granite, (**g**) Rosa granite, (**h**) Qusseir brown.

**Figure 2 materials-15-02041-f002:**
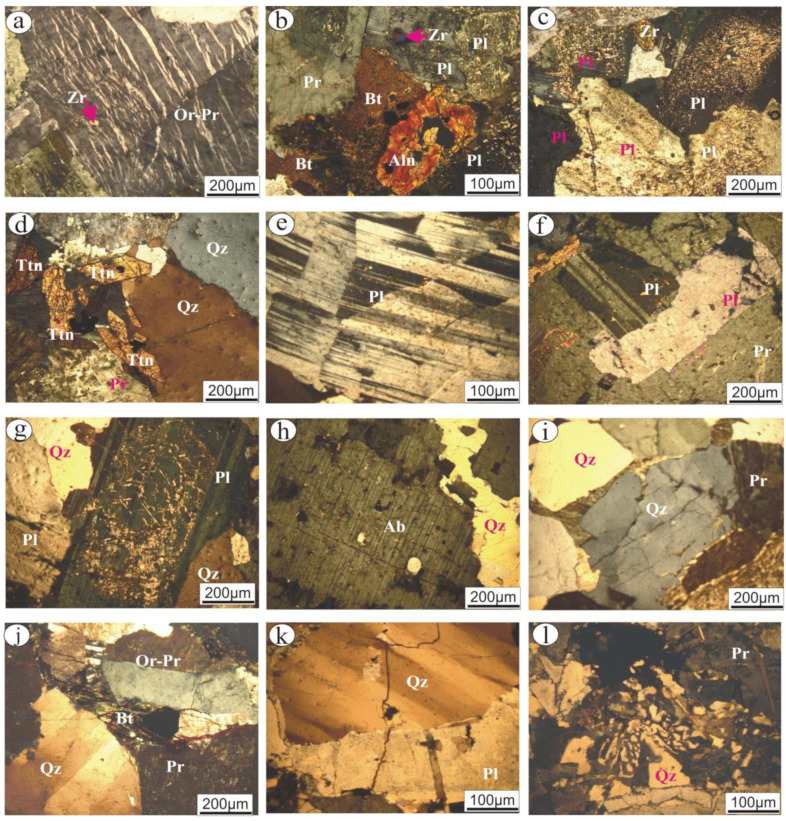
Photomicrographs of the examined commercial granitic rocks showing: Gandonna rock sample: (**a**) Simple twining of flamy orthoclase perthite (Or-Pr) engulfing minute zircon (Zr) crystal; (**b**) Allanite (Aln) (filled by iron oxides) associated with biotite (Bt). Pink granite rock sample: (**c**) Aggregate of turbid plagioclase (Pl) crystals due to extensive saussuritization; (**d**) Euhedraltitanite (Tnt) in association with quartz (Qz). Buff granite rock sample I: (**e**) Slightly saussuritized twisted plagioclase. Buff granite rock sample II: (**f**) Pristine perthite crystal engulfing plagioclase and biotite; (**g**) Extensive saussuritization of plagioclase associated with quartz. Fantazia rock sample: (**h**) Microveinlets of secondary quartz-traversed coarse-grained perthite crystals. Red granite rock sample: (**i**) Reaction rim of albite developed around kaolinitizedperthite; and (**j**) Simple twinning of orthoclase perthite associated with undulose quartz. Qusseir brown rock sample: (**k**) Extended fracture in perfect undulose quartz and Kaolinitized plagioclase filled by secondary quartz; (**l**) Perfect granophyric texture.

**Figure 3 materials-15-02041-f003:**
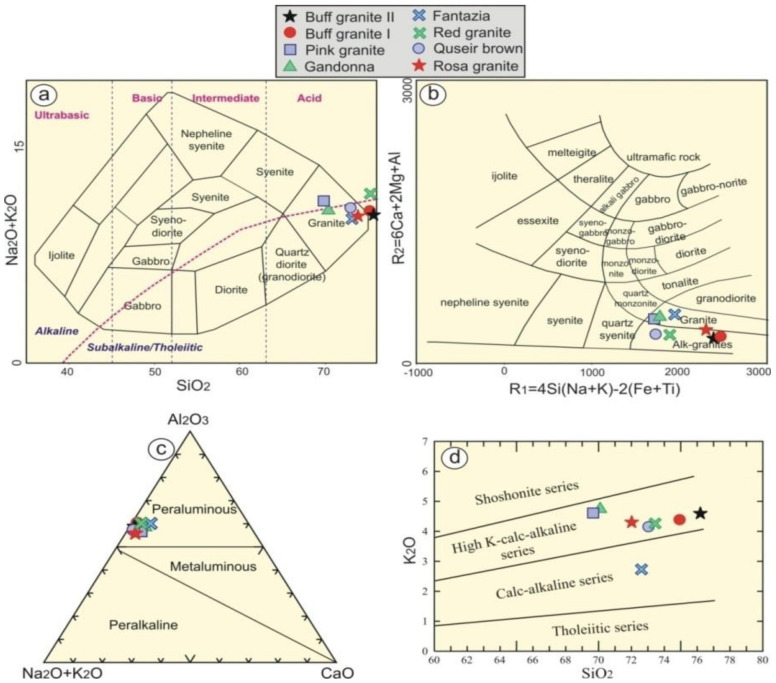
Classification of the studied granitic rocks using (**a**) SiO_2_ vs. Na_2_O + K_2_O diagram after Cox et al. [24]; (**b**) the R1-R2 diagram (De la Roche et al.) [25]. Magma type diagrams of (**c**) CaO-(Na_2_O + K_2_O)-Al_2_O_3_ discrimination diagram (Shand) [26] and (**d**) SiO_2_ vs. K_2_O diagram (Rickwood) [27].

**Figure 4 materials-15-02041-f004:**
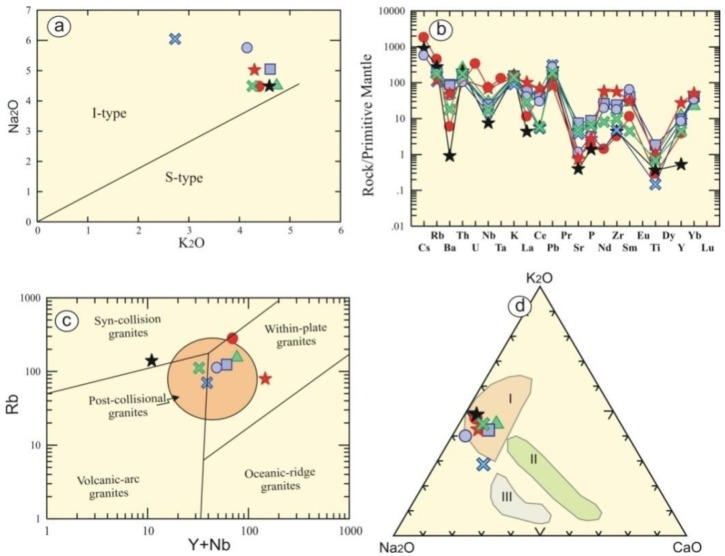
(**a**) K_2_O-Na_2_O binary diagram (Hine et al.) [28]; (**b**) Trace elements pattern normalized to the primitive mantle (Sun and McDonough) [29]. Tectonic discrimination diagrams for (**c**) Rb vs. Y + Nb binary diagram (Pearce et al.) [30], where the post-collisional field is adapted from (Pearce) [31]; and (**d**) Na_2_O-K_2_O-CaO of the Egyptian granitoids (Hassan and Hashad) [32]. I = late (subphase) orogenic calc-alkaline phase, II = early subphase of younger granites, III = older calc-alkaline granites.

**Figure 5 materials-15-02041-f005:**
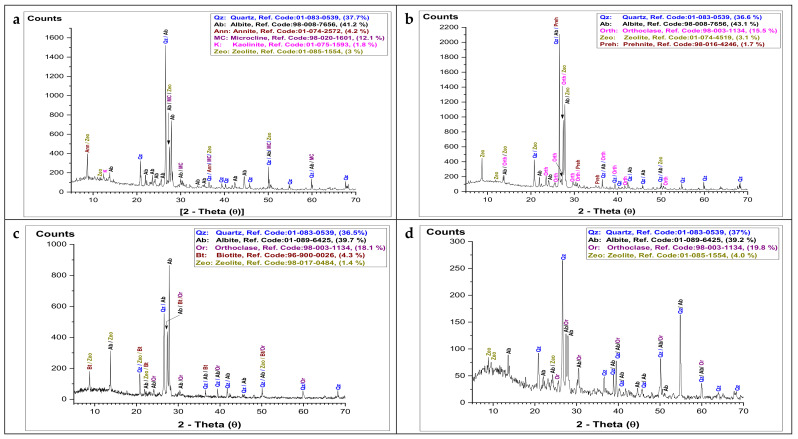
X-ray diffraction (XRD) patterns showing the main mineralogical composition of different granitic rocks: (**a**) Gandonna; (**b**) pink granite; (**c**) buff graniteI; (**d**) buff graniteII; (**e**) Fantazia; (**f**) red granite; (**g**) Rosa granite; (**h**) Qusseir brown.

**Figure 6 materials-15-02041-f006:**
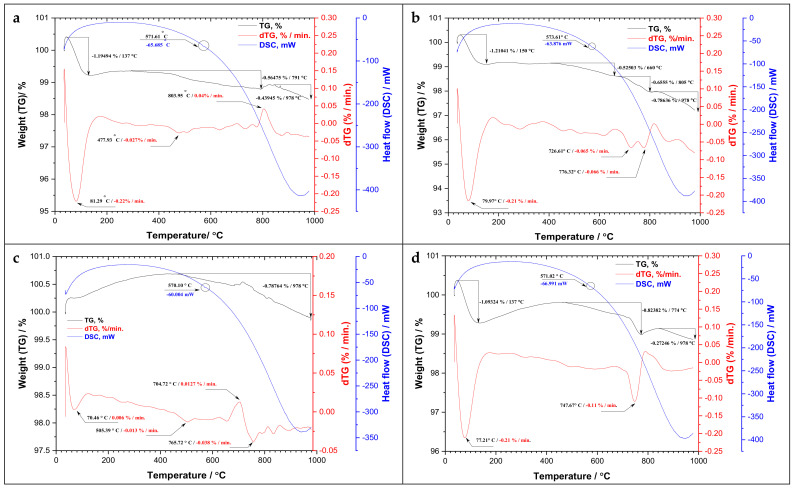
Thermogravimetric analysis of different granitic rocks using TG/DTG/DSC: (**a**) gandonna; (**b**) pink granite; (**c**) buff graniteI; (**d**) buff graniteII; (**e**) fantazia; (**f**) red granite (**g**) rosa granite; (**h**) qusseir brown. A circle shape represents the phase transition of α-β quartz at 573 °C.

**Figure 7 materials-15-02041-f007:**
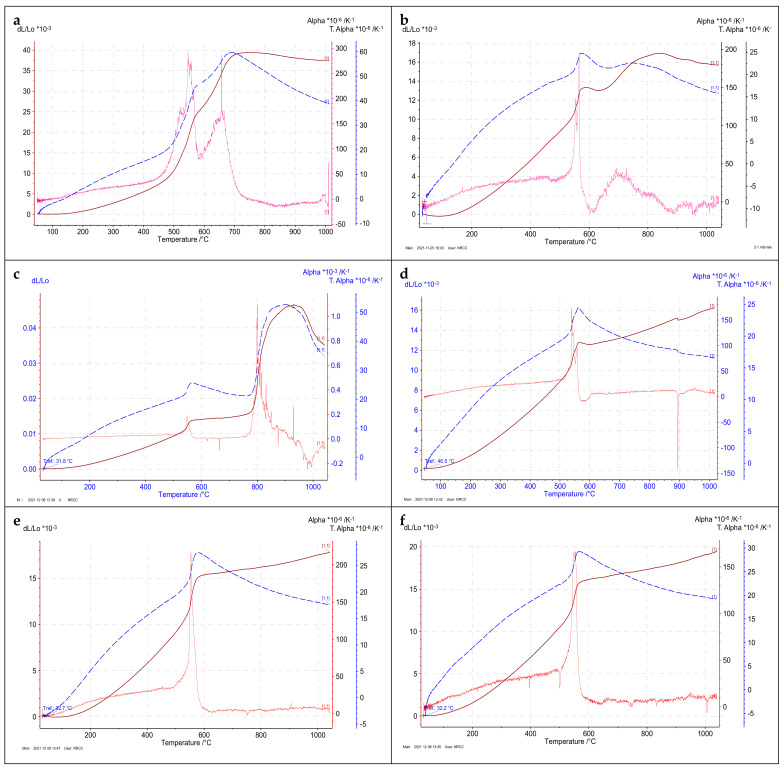
Linear thermal expansion (LTE) patterns showing the behavior of different granitic rocks under increasing temperatures: (**a**) gandonna; (**b**) pink granite; (**c**) buff granite I; (**d**) buff granite II; (**e**) fantazia; (**f**) red granite; (**g**) rosa granite; (**h**) qusseir brown.

**Figure 8 materials-15-02041-f008:**
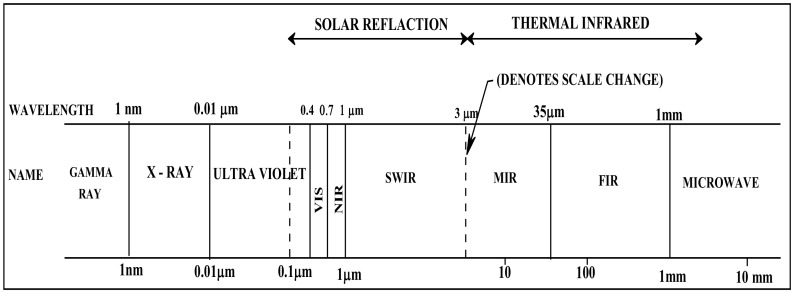
Different parts of electromagnetic (EM) spectrum.

**Figure 9 materials-15-02041-f009:**
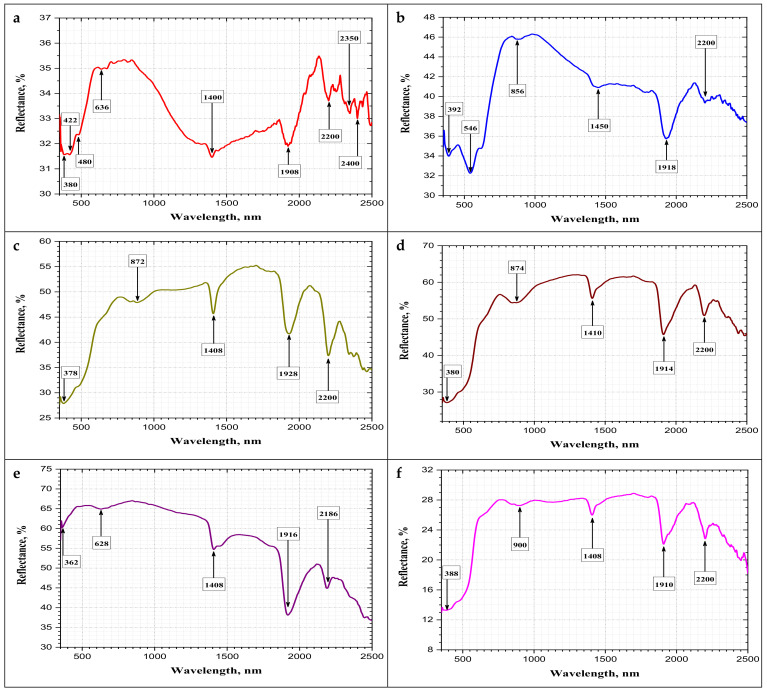
Spectral reflectance patterns in VIS-NIR and SWIR regions (350–2500 nm) of different granitic rocks: (**a**) gandonna; (**b**) pink granite; (**c**) buff granite I; (**d**) buff granite II; (**e**) fantazia; (**f**) red granite; (**g**) rosa granite; (**h**) qusseir brown.

**Figure 10 materials-15-02041-f010:**
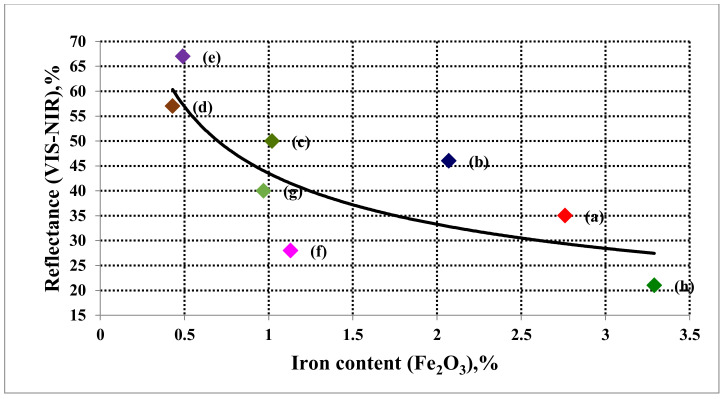
Relationship between iron content and reflectance of igneous rock types in the VIS-NIR region (400–1000 nm).

**Figure 11 materials-15-02041-f011:**
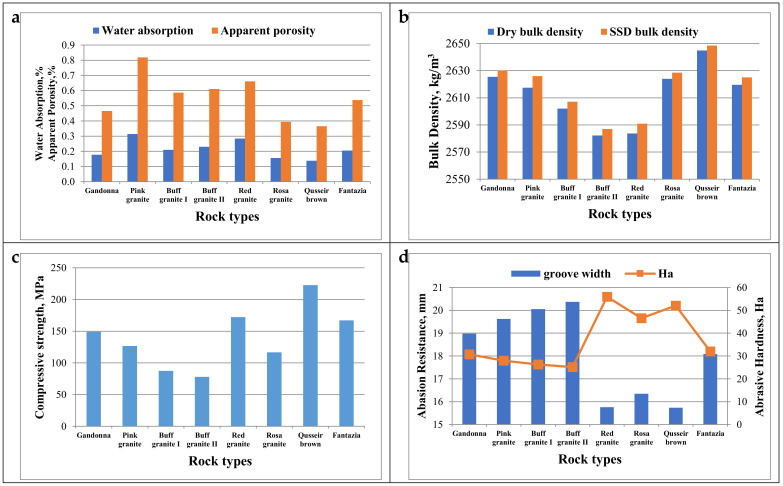
Physico-mechanical properties of different granitic rock types. (**a**) water absorption and porosity; (**b**) dry and saturated surface dry (SSD) bulk density; (**c**) compressive strength; (**d**) abrasion resistance.

**Table 1 materials-15-02041-t001:** Summary of petrographical investigation of granitic rocks.

Rock Types			Petrographical Properties							
		Rock Classification	Mineral Composition							
			Primary				Accessory		Secondary	
a	Gandonna	Monzogranite	Plagioclase	K-feldspar	Quartz	Biotite	Allanite	Zircone	(Chlorite)	
b	Pink granite	Monzogranite	Plagioclase	K-feldspar	Quartz	Biotite	Titanite	Zircone	(Chlorite)	
c	Buff granite-I	Syanogranite	K-feldspar	Plagioclase	Quartz	Muscovite			(Saussurite)	(Kaolinite)
d	Buff granite-II	Syanogranite	K-feldspar	Plagioclase	Quartz	Muscovite			(Saussurite)	
e	Fantazia	Monzogranite	Plagioclase	Quartz	(K-feldspar)	Biotite			(Saussurite)	
f	Red granite	Syanogranite	K-feldspar	Plagioclase	Quartz	Biotite			(Kaolinite)	(Saussurite)
g	Rosa granite	Monzogranite	K-feldspar	Plagioclase	Quartz	Biotite			(Chlorite)	
h	Qusseir brown	Monzogranite	K-feldspar	Plagioclase	Quartz	Biotite			(Kaolinite)	(Saussurite)

**Table 2 materials-15-02041-t002:** Chemical composition [major (%) and trace elements (ppm)] of the studied granitic rock types.

Oxides	Gandonna	Pink Granite	Buff Granite (I)	Buff Granite (II)	Fantazia	Red Granite	Rosa Granite	Qusseir Brown
SiO_2_	70.097	69.673	74.931	76.197	72.61	73.035	73.444	72.018
Al_2_O_3_	14.954	14.28	14.127	13.487	15.765	13.933	14.397	12.747
TiO_2_	0.215	0.397	0.062	0.079	0.032	0.155	0.15	0.209
Fe_2_O_3_	2.76	2.066	1.018	0.544	0.486	1.126	0.974	3.294
MgO	0.503	0.903	0.2	0.139	0.163	0.299	0.51	0.183
Na_2_O	4.503	5.049	4.482	4.488	6.046	5.757	4.489	5.024
K_2_O	4.739	4.61	4.388	4.595	2.729	4.153	4.261	4.3
CaO	1.352	1.216	0.42	0.386	1.693	0.469	0.776	0.768
MnO	0.059	0.08	0.133	0.01	0.053	0.14	0.065	0.101
P_2_O_5_	0.146	0.19	0.043	0.03	0.073	0.073	0.137	0.057
LOI	0.46	0.44	0.41	0.16	0.23	0.67	0.604	0.72
SC	5	UDL	10.3	3	UDL	5.7	UDL	UDL
V	UDL	UDL	2.9	2.2	UDL	UDL	UDL	UDL
Co	19.1	44.5	UDL	2.3	21.2	29.8	UDL	28.4
Ni	2.3	13.2	2.1	2.5	10.3	9.1	UDL	9.4
Cu	UDL	UDL	4.5	4.2	UDL	UDL	3.4	UDL
Zn	75.2	69.7	93.2	34.7	25.3	56.1	36.1	144
Ga	21.7	19.4	30.9	22.7	25.9	17..9	16.6	28.4
Ge	4.9	7.7	2.3	UDL	7.6	6.3	UDL	5.3
Rb	154.3	124	282	175.4	70.2	112.7	111.5	79.5
Sr	132.7	158	18.9	8.4	79.5	24.5	101.1	15.7
Y	56.1	43.5	17.3	2.4	38.9	38.1	20.2	124.9
Zr	192.8	268.3	36	49.4	53.7	195	107.8	627.3
Nb	20.8	17.3	52.1	5.4	UDL	10.3	12.2	49.4
Mo	UDL	3.3	UDL	UDL	UDL	UDL	UDL	4.6
Ag	12.2	8.2	UDL	UDL	19.2	15.4	UDL	10.7
Cd	9.7	UDL	UDL	UDL	12	6.1	UDL	0.8
Sn	9	6.3	11.9	5.6	9.8	5.9	13.2	12.3
I	10.9	12.8	UDL	UDL	6.6	8	UDL	12.1
Cs	UDL	UDL	14.2	7.2	UDL	4.5	UDL	UDL
Ba	322.9	620.9	41.4	6.4	UDL	305.9	125.8	335.1
La	32.2	40.2	7.8	3	UDL	27.7	19.5	69.8
Ce	113.3	70.4	0	10.3	9.4	52.7	10.4	122.6
Nd	48.1	35.4	1.9	UDL	UDL	26.2	10.9	78.3
Sm	12.6	18	5	UDL	UDL	28.5	2	13.8
Yb	11	20.5	UDL	UDL	19.3	16.7	UDL	24.8
Hf	UDL	UDL	5.2	UDL	UDL	UDL	5.5	3.7
Ta	UDL	UDL	5.3	UDL	UDL	UDL	UDL	UDL
W	297.5	556.9	6	17	479.8	457.7	5.3	410.9
Hg	180.7	374.3	UDL	UDL	313.6	279.1	UDL	268.4
Tl	UDL	UDL	3.9	5.3	UDL	UDL	8.2	UDL
Pb	19.8	12.6	9.9	17.2	21.8	19.2	13.3	5.7
Th	21.4	14.4	9	17.9	UDL	8.7	15	UDL
U	UDL	UDL	7.2	UDL	UDL	UDL	UDL	UDL

**Table 3 materials-15-02041-t003:** Quantitative mineral composition of granitic rock types using Rietveld refinement XRD.

Rock Types		Mineral Composition,%						
		Quartz	Plagioclase	Alkali Feldspar	Mica	Alteration Minerals		
			Albite	Orthoclase	Annite/Biotite	Zeolite	Prehnite	Kaolinite
a	Gandonna	37.7	41.2	12.1	4.2	3	---	1.8
b	Pink granite	36.6	43.1	15.5	---	3.1	1.7	---
c	Buff graniteI	36.5	39.7	18.1	4.3	1.4	---	---
d	Buff graniteII	37	39.2	19.8	---	4	---	---
e	Fantazia	32.1	58.9	9	---	---	---	---
f	Red granite	24.5	47.2	28.3	---	---	---	---
g	Rosa granite	34.3	40.3	25.4	---	---	---	---
h	Qusseir brown	31.3	41.1	26.3	---	---	1.3	---

**Table 4 materials-15-02041-t004:** Measurements of (*dL*/*Lo*, %) and (*α_T_*, 10^−6^/K^−1^) for granitic rock types at different temperatures.

Temperature, (T), °C	Granitic Rock Types															
	Gandonna		Pink Granite		Buff GraniteI		Buff GraniteII		Fantazia		Red Granite		Rosa Granite		Quessir Granite	
	*dL*/*Lo*, %	*α_T_*	*dL*/*Lo*, %	*α_T_*	*dL*/*Lo*, %	*α_T_*	*dL*/*Lo*, %	*α_T_*	*dL*/*Lo*, %	*α_T_*	*dL*/*Lo*, %	*α_T_*	*dL*/*Lo*, %	*α_T_*	*dL*/*Lo*, %	*α_T_*
**50**	0.0140	26.35	−0.010	−5.06	0.005	2.85	0.014	23.22	0.0014	0.69	0.0048	2.44	−0.013	−6.4	0.0081	4.42
**100**	0.0066	1.28	−0.0190	−2.73	0.0241	3.47	0.0318	5.68	−0.0007	−0.02	0.02582	3.75	−0.015	−2.3	0.0245	3.52
**300**	0.2758	10.97	0.2940	10.95	0.3520	13.11	0.3456	13.58	0.3051	11.34	0.3956	14.7	0.3827	14.2	0.3135	11.7
**500**	1.0751	23.78	0.8710	18.54	0.9381	20.05	0.8913	19.54	0.9116	19.49	1.0503	22.5	1.2811	27	0.8703	18.6
**700**	3.8856	59.70	1.4433	21.61	1.4656	21.82	1.3183	20.10	1.5800	23.63	1.6765	25.1	9.010	134	1.5341	23
**1000**	3.7579	39.39	1.5835	16.33	3.8806	40.04	1.608	16.83	1.7534	18.10	1.9067	19.7	5.3443	55.1	1.7391	17.9

## Data Availability

The study did not report any data.

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
