# Peer review of "Relationship of Mineralogical Composition to Thermal Expansion, Spectral Reflectance, and Physico-Mechanical Aspects of Commercial Ornamental Granitic Rocks"

_materials, 2022, doi:10.3390/ma15062041_

Round 1

Reviewer 1 Report

The paper is well structured, and several tests have been carried out to test the properties of some granite rocks, widely used in architectural contexts.

Just a few small tips:

I suggest a summary table to enclose the mineral-petrographic properties. The table would help the reader in a more fluent reading where all the properties of the samples observed in thin section and analyzed with XRD are summarized. The following papers should be cited in the introduction when talking about the materials and tests to be conducted, especially when talking about materials used in the construction and building materials field: “The contribution of earth sciences to the preservation of underwater archaeological stone materials: An analytical approach. International Journal of Conservation Science, Volume 6, Issue 3, Pages 335 - 348, 2015”; "Modelling the compressive mechanical behaviour of granite and sandstone historical building stones. https://doi.org/10.1016/j.conbuildmat.2011.08.083"

Fig. 1 Please add a bar scale.

Fig. 2 bar scale is missing in images b and c.

After these small changes and additions, the paper can be published.

Author Response

Response to Reviewer 1 Comments

Point 1:

I suggest a summary table to enclose the mineral-petrographic properties. The table would help the reader in a more fluent reading where all the properties of the samples observed in thin section and analyzed with XRD are summarized.

Response 1:

A table summarizing the petrographical investigation of the studied granitic rock types was added.

Point 2:

The following papers should be cited in the introduction when talking about the materials and tests to be conducted, especially when talking about materials used in the construction and building materials field: “The contribution of earth sciences to the preservation of underwater archaeological stone materials: An analytical approach. International Journal of Conservation Science, Volume 6, Issue 3, Pages 335 - 348, 2015”; "Modelling the compressive mechanical behaviour of granite and sandstone historical building stones. https://doi.org/10.1016/j.conbuildmat.2011.08.083"

Response 2:

The referred articles were cited in the introduction section [6, 24].

Point 3:

Fig. 1 Please add a bar scale.

Response 3:

The bar scale for the referred Fig was added.

Point 4:

Fig. 2 bar scale is missing in images b and c.

Response 4:

The missing bar scale for the referred images was added.

Reviewer 2 Report

Author have tried to study the thermal expansion, spectral reflectance, and physico- mechanical aspects of different types of commercial granitic rocks with their mineralogical and chemical composition.

English need to be studied

Significance of current work need to be described in detail

The results need to be compared with other researchers as well.

Some of the recent references need to be included like

PK Gautam, MK Jha, AK Verma, TN Singh (2019) Evolution of absorption energy per unit thickness of damaged sandstone, Journal of Thermal Analysis and Calorimetry 136 (6), 2305-2318

Author Response

Response to Reviewer 2 Comments

Point 1:

 English need to be studied

Response 1:

The english has been revised

Point 2:

Significance of current work need to be described in detail

Response 2:

The significance of the present study was decribed in detail under the title “Research significance” in the introduction section.

Point 3:

The results need to be compared with other researchers as well.

Response 3:

                The test results were compared with other cited researchers.

Point 4:

Some of the recent references need to be included like:

PK Gautam, MK Jha, AK Verma, TN Singh (2019) Evolution of absorption energy per unit thickness of damaged sandstone, Journal of Thermal Analysis and Calorimetry 136 (6), 2305-2318

Response 4:

                The referred reference was included in the manuscript [40].

Reviewer 3 Report

  1. Title too long and not reflected to content of study. Word "Some" also not suitable used.
  2. Abstract lacking an information. Background study, problem statement and points out research gaps, objectives/aims, summary of methods, and novelty of research study are not clearly presented.
  3. "The present work is a comprehensive study that aims at linking the variations..". This sentences doesn't have correlation with aim of study.
  4. Page 3 line 112-119. Methodology must be place in other method section, not under introduction.
  5. Physical properties of different granitic rock is compulsory to include in paper. As aim of study was about Commercial Granitic Rocks.
  6. "Petrographical investigation". Details description is required. Not only presenting the objective of test.
  7. For TG test, why use  temp. range form 34 to 978 0C? More explanation is needed.

Author Response

Response to Reviewer 3 Comments

Point 1:

Title too long and not reflected to content of study. Word "Some" also not suitable used.

Response 1:

The manuscript title has been modified to reflect the aim of study.

Point 2:

Abstract lacking an information. Background study, problem statement and points out research gaps, objectives/aims, summary of methods, and novelty of research study are not clearly presented.

Response 2:

All items are modified.

Point 3:

"The present work is a comprehensive study that aims at linking the variations..". This sentences doesn't have correlation with aim of study.

Response 3:

The aim of the present study is to link the variations in various aspects of granitic rocks to their mineralogical and chemical composition. Therefore, this study included the effect of mineral composition on several characteristics of commercial granitic rocks.

Point 4:

Page 3 line 112-119. Methodology must be place in other method section, not under introduction.

Response 4:

The lines 122-119 have been moved and placed under the subtitle 2.2 “Methods”.

Point 5:

Physical properties of different granitic rock is compulsory to include in paper. As aim of study was about Commercial Granitic Rocks.

Response 5:

The physical properties of the studied commercial granitic rocks have already been included in the present study including “water absorption, bulk density (dry&ssd), and apparent porosity”.

Point 6:

"Petrographical investigation". Details description is required. Not only presenting the objective of test.

Response 6:

The studied granitic rock types were microscopically described in detail under the subtitle 3.1 “Petrographical investiagtion”.

Point 7:

For TG test, why use  temp. range form 34 to 978 °C? More explanation is needed

Response 7:

The starting temperature of TG test (34 °C) is the room temperature. The ending temperature was planned to reach 1000 °C, but as a result of no further change in the thermal curves, the test was stopped at 978 °C.  

Reviewer 4 Report

The manuscript is recommended for the publication considering the following comments.

-The introduction (page 2) includes very generic information on the history and place of the production, which seem to be out of scope of this manuscript. I would recommend to remove/shorten this part and put the focus on the material/ engineering properties and literature of the scope of this study.

-Looking to literature review, it lacks information on the gab of the previous studies in which this manuscript could fill this/these gab/s.

-Section 2, provides short/brief information on the performed tests while Section 3 provides the results. However detailed description on the procedures of testing is missing. It is highly recommended to insert the method of experimental measurements including dimension of sample, how to calculate the percentages of chemical compositions, number of specimens, etc.

Author Response

Response to Reviewer 4 Comments

Point 1:

The introduction (page 2) includes very generic information on the history and place of the production, which seem to be out of scope of this manuscript. I would recommend to remove/shorten this part and put the focus on the material/ engineering properties and literature of the scope of this study.

Response 1:

Your recommendation has been taken in consideration and the information concerning the place of production of granitic rocks are removed, while that concerning the geology of granitic rocks in Egypt has been shortened.

Point 2:

Looking to literature review, it lacks information on the gab of the previous studies in which this manuscript could fill this/these gab/s.

Response 2:

From the literature review, there are not enough studies about the physical and mechanical properties of granitic igneous rock and their relationship to the mineralogical composition and their suitability as ornamental stones.

Point 3:

Section 2, provides short/brief information on the performed tests while Section 3 provides the results. However detailed description on the procedures of testing is missing. It is highly recommended to insert the method of experimental measurements including dimension of sample, how to calculate the percentages of chemical compositions, number of specimens, etc.

Response 3:

Your recommendation has been taken in consideration but the sample dimensions have already been mentioned for each test in the manuscript. Regarding the number sample, it was added. The calculation of chemical composition was automatically conducted using XRF at National Research Centre Laboratories (Axios PANalytical 2005, Sequential WD- XRF Spectrotometer using ASTM E-1621 standard guide for elemental analysis by wavelength-dispersive X-ray fluorescence Spectrometer). The data were processed through advanced data treatment software of PAnalytical Super Q package with the multi-elemental synthetic standards prepared by BGS/PAnalytical Corporation

Round 2

Reviewer 3 Report

The revised manuscript is acceptable.

Reviewer 4 Report

The manuscript has been revised. It is recommended for the publication.